# Improving Adversarial Robustness via Information Bottleneck Distillation

**Huafeng Kuang**[1], **Hong Liu**[2]*, **YongJian Wu**[3], **Shin'ichi Satoh**[2], **Rongrong Ji**[1]†

[1]Key Laboratory of Multimedia Trusted Perception and Efficient Computing,
Ministry of Education of China, Xiamen University, 361005, P.R. China
[2]National Institute of Informatics, Tokyo, 101-8430, Japan
[3]Youtu Laboratory, Tencent, Shanghai, 200233, China
`skykuang@stu.xmu.edu.cn, lynnliu.xmu@gmail.com`
`littlekenwu@tencent.com, satoh@nii.ac.jp, rrji@xmu.edu.cn`

## Abstract

Previous studies have shown that optimizing the information bottleneck can significantly improve the robustness of deep neural networks. Our study closely examines the information bottleneck principle and proposes an Information Bottleneck Distillation approach. This specially designed, robust distillation technique utilizes prior knowledge obtained from a robust pre-trained model to boost information bottlenecks. Specifically, we propose two distillation strategies that align with the two optimization processes of the information bottleneck. Firstly, we use a robust soft-label distillation method to increase the mutual information between latent features and output prediction. Secondly, we introduce an adaptive feature distillation method that automatically transfers relevant knowledge from the teacher model to the student model, thereby reducing the mutual information between the input and latent features. We conduct extensive experiments to evaluate our approach's robustness against state-of-the-art adversarial attackers such as PGD-attack and AutoAttack. Our experimental results demonstrate the effectiveness of our approach in significantly improving adversarial robustness. Our code is available at https://github.com/SkyKuang/IBD.

## 1 Introduction

Numerous works have shown that deep neural networks (DNNs) are easily attacked by adversarial examples [56, 41, 4, 14], which involve adding imperceptible noise to inputs and causing incorrect outputs. This vulnerability of DNNs raises significant security concerns when deploying DNNs in safety-critical applications. To address this potential threat, various adversarial defense strategies have been proposed [44, 41, 72, 43, 72]. Among these defenses, adversarial training (AT) [41, 71] is a general solution for defending against adversarial attacks by incorporating adversarial examples generated by an adversarial attack into the training process. While many AT techniques can defend against sophisticated attacks, such as AutoAttack [14], a big robust generalization gap still exists between the training data and testing data [12].

Recent years have witnessed the growing popularity of the studies of the use of **I**nformation **B**ottleneck (IB) [57] in training robust DNNs [54, 2, 64, 68]. IB involves finding a trade-off in intermediate features $Z$ between relevant information for the prediction $Y$ and nuisance information about input $X$. The overall objective of IB is formulated as follows:

$$\max I(Z;Y) - \beta I(X;Z), \tag{1}$$

---

*Project lead
†Corresponding author

37th Conference on Neural Information Processing Systems (NeurIPS 2023).

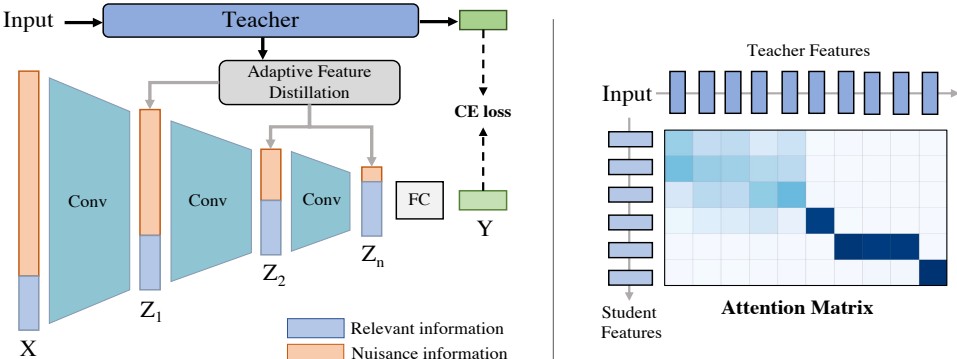

Figure 1: Schematic of successive extraction of relevant information in IBD. The inputs are passed through a robust teacher and a student to get intermediate features (Z). These features are then passed through an adaptive feature distillation module, where an attention-based model determines similarities between the teacher and student features. Robust information from each teacher feature is transferred to the student with the attention values. The student model also minimizes cross-entropy (CE) loss using the teacher's soft output label. Consult the section 3.2 for more details.

where $I$ denotes mutual information and $\beta$ controls the trade-off between the two terms. Studies in [54, 64] show that IB can produce concise representations leading to better generalization, and can make intermediate features of DNNs less sensitive to input perturbations. On the other hand, variational information bottleneck (VIB) has also been found to significantly improve the robustness of models [2, 19]. In addition, [40] replaces mutual information with the Hilbert Schmidt Independence Criterion (HSIC) and calls this the HSIC Bottleneck, and [64] further investigates the HSIC bottleneck as a regularization to enhance the robustness of DNNs.

The crucial step to optimizing the IB is to calculate mutual information in an efficient and effective manner. An approach in [19] proposes a conditional entropy bottleneck (CEB) (see Eq.(6)), which enhances IB by introducing label priors in variational inference. This approach employs a backward encoder to approximate the true distribution $p(z|y)$. However, DNN processing can form a Markov chain $(X \rightarrow Z \rightarrow Y)$ and obey the data processing inequality: $H(X) \geq H(Z) \geq H(Y)$, and the label $Y$ often cannot satisfy the intrinsic dimension requirement [37]. As a result, directly using label information to approximate $p(z|y)$ is imprecise. Furthermore, the backward encoder usually provides uniform priors for inputs with the same label and cannot provide customized priors based on different inputs, particularly for high-dimensional image data.

Fortunately, prior research has shown that robust pre-trained models can provide robust prior information to enhance model robustness and certainty [27, 53, 21]. Motivated by this, we revisit VIB through the lens of robustness distillation and further introduce a new IB objective: **I**nformation **B**ottleneck **D**istillation (IBD) (see Eq.(7)). To make IBD practical, we put forward two distillation strategies meant to target the two optimization processes of the information bottleneck respectively. Firstly, we utilize soft-label distillation to maximize the mutual information between intermediate features and output prediction. Secondly, we use adaptive feature distillation to restrict the mutual information between the input and intermediate features, which facilitates the transfer of appropriate knowledge from the teacher model to the student model, ultimately resulting in a more accurate estimation of the distribution of student features. A schematic diagram illustrating the IBD is shown in Figure 1.

Overall, we make the following contributions:

- **Theoretically**, we utilize conditional variational inference to construct a lower bound to estimate the mutual information and reformat the IB principle by using the adversarial robustness as the prior for learning features, which is termed Information Bottleneck Distillation (IBD).

- **Algorithmically**, to realize IBD, we propose two distillation strategies: robust soft-label and adaptive feature distillation, to match the two optimization processes of the information bottleneck, respectively.

- **Experimentally**, we conducted extensive experiments on various benchmark datasets such as CIFAR and ImageNet. The results show the effectiveness of our IBD in improving the robustness of DNNs against most attacks (e.g., PGD-attack and AutoAttack), and our IBD behaves more robustly than state-of-the-art methods.

## 2  Background & Related Work

**Adversarial Attacks.** Adversarial examples are observed when an attacker adds human-imperceptible perturbation to the inputs, which misleads DNN's output [56]. And a series of attacking algorithms are proposed [22, 4, 41, 18, 14, 38]. One of the most popular attacks is the Project Gradient Descent (PGD) attack [41], which uses multi-step projected gradient descent to generate adversarial examples:

$$x_k = \Pi\Big(x_{k-1} + \alpha \cdot \text{sign}\big(\nabla_x \mathcal{L}_{\mathcal{CE}}(h_\theta(x_{k-1}), y)\big)\Big), \tag{2}$$

where $\alpha$ is the step size, $\Pi$ is the project function, and $x_k$ is the adversarial example at the $k$-th step, $h_\theta$ is a DNN model with weight $\theta$, and $\mathcal{L}_{CE}$ is cross-entropy function. Another widely-used adversarial attack method is the C&W attack [4], which applies a rectifier function regularizer to generate adversarial examples near the original input. AutoAttack (AA) [14], an ensemble of different parameter-free attacks, is currently regarded as the most reliable evaluation of the adversarial robustness of the model. Black-box attacks, which do not require full knowledge of DNNs, are more practical, and previous works [67, 18] have shown that adversarial examples generated by one model can attack other models with a high probability. Query-based black-box attacks [3, 9] update the perturbation iteratively according to the attack objective. In this work, we test the robust performance under both white- and black-box attacks to verify the robustness of our proposed method.

**Adversarial Robustness.** To counter the threat of adversarial attacks, researchers have proposed various defense methods, including defensive distillation [44], manifold-projection [51], pre-processing [24, 69], provable defenses [45, 50], and adversarial training [11, 22, 41]. Among these, adversarial training **(AT)** [41] is a highly effective and widely-used solution. It involves adding adversarial examples, generated by the adversarial attack scheme, to the training data, which can be formulated by a min-max optimization problem:

$$\min_\theta \mathbb{E}\Big[ \max_{x_{\text{adv}}} \mathcal{L}_{CE}\big(h_\theta(x_{\text{adv}}), y\big) \Big]. \tag{3}$$

where $x_{\text{adv}}$ is adversarial example generated by a special attack algorithm. Many variants of AT have been proposed. [71] proposed the TRADES, which characterizes the trade-off between accuracy and robustness. [66] proposed adversarial weight perturbation (AWP), which uses weight perturbation to increase the robustness of the model. In addition, some new efforts have also been devoted from different aspects including designing new training strategies [43, 33], adversarial regularization [42, 48, 39], robustness architecture search [8, 25, 29, 30], and data augmentation [5, 46, 63].

**Robustness Distillation.** Knowledge distillation (KD) [28] is an effective method, wherein a well-trained teacher model is used to supervise the learning of the student model. The overall objective of KD is formulated as:

$$\min_{\theta_S} \mathbb{E}\Big[ \mathcal{L}_{task}\big(h_s(x), y\big) + \alpha \cdot L_{Dist}\big(h_s(x), h_t(x)\big) \Big], \tag{4}$$

where $\mathcal{L}_{task}$ is the task-specific loss function for the target task and $\mathcal{L}_{Dist}$ is the loss function that penalizes the difference between the teacher and the student. Recent studies show that KD can also be used to enhance the adversarial robustness of a student network [44]. Then, the adversarially robust distillation (ARD) method, as introduced in [21], combines adversarial training with KD to further increase the robustness of the student model under the supervision of an adversarially robust teacher network. Introspective Adversarial Distillation (IAD) [73] propose proposes a method to automatically realize the intuition of the previous reliable guidance during the adversarial distillation. And, RSLAD [74] improves the performance of ARD by using robust soft labels generated by a pre-trained teacher network to replace hard labels used in [21].

# 3 Information Bottleneck Distillation

In this section, we revisit the concept of information bottleneck from the perspective of robust distillation and establish the IBD objective. We further propose two distillation strategies to perform the two optimization processes of the information bottleneck, respectively. Finally, we combine IBD with AT to learn a robust model.

## 3.1 A Distillation View of Information Bottleneck

The information bottleneck theory has been used to explain deep models [58], compress models [16], and learn robust models [2]. To make the IB practical, a major challenge is to accurately estimate the mutual information. In [2], Variational Information Bottleneck (VIB)[1] utilizes variational inference to construct the lower bound to estimate the mutual information. VIB is formulated as:

$$I(Z;Y) - \beta I(X;Z) \geq \mathop{\mathbb{E}}_{p(x,y)p(z|x)} \left[ \log q(y|z) - \beta \log \frac{p(z|x)}{q(z)} \right], \tag{5}$$

where $p(z|x)$ is feature distribution, $q(y|z)$ is a variational approximation to the true distribution $p(y|z)$, and $q(z)$ is the variational approximation to the true distribution $p(z)$. To optimize Eq.(5), VIB uses neural networks to parameterize the Gaussian densities $p(z|x)$ and $q(y|z)$, and treats $q(z)$ as a fixed $K$-dimensional spherical Gaussian distribution $q(z) = \mathcal{N}(0, I)$ (where $K$ is the size of the bottleneck). Then, the reparameterization trick and Monte Carlo sampling are used to get an unbiased estimate of the gradient. This process thus allows DNNs to handle high-dimensional data, such as images, avoiding previous limitations on discrete or Gaussian distribution cases.

Next, how to determine the approximate $q(z)$ is important in variational inference. VIB empirically treats $q(z)$ as a fixed Gaussian distribution. For a better approximation of $q(z)$, [19] proposes the Conditional Entropy Bottleneck (CEB), which introduces label prior information to approximate $q(z)$. The CEB can be formulated as:

$$I(Z;Y) - \beta I(X;Z|Y) \geq \mathop{\mathbb{E}}_{p(x,y)p(z|x)} \left[ \log q(y|z) - \beta \log \frac{p(z|x)}{q(z|y)} \right]. \tag{6}$$

In detail, CEB utilizes a linear mapping layer to parameterize $q(z|y)$ that takes a one-hot label $y$ as input and outputs a vector $\mu_y$ as the mean of the Gaussian distribution $q(z|y) = \mathcal{N}(\mu_y, I)$. CEB also uses an identity matrix $I$ for the variance of both $q(z|y)$ and $p(z|x)$. Note that CEB and VIB differ in terms of the presence of a class conditional or unconditional variational marginal, and whether $q(z|y)$ and $p(z|x)$ have a fixed variance. Thus, by introducing such a class prior, CBE can learn an approximate distribution better. However, using the one-hot label alone to approximate high-dimensional distribution $q(z)$ is inaccurate enough due to data processing inequality and intrinsic dimension [37], particularly for high-dimensional image data.

To address this challenge, we propose leveraging adversarial robustness distillation [21] to build the information bottleneck objective. This approach involves providing robust prior information obtained from a robust pre-trained model, which aims to improve model robustness and reduce model uncertainty [27, 53]. Specifically, we replace conventional one-hot label prior information with robust prior information derived from such a robust pre-trained model. We call this method **I**nformation **B**ottleneck **D**istillation (IBD). To implement IBD, we approximate $q(z)$ by utilizing the intermediate features extracted by an adversarially pre-trained model, and our final objective is formulated as:

$$I(Z;Y) - \beta I(X;Z|T) \geq \mathop{\mathbb{E}}_{p(x,y)p(z|x)} \left[ \log q(y|z) - \beta \log \frac{p(z|x)}{q(z|t)} \right], \tag{7}$$

where $T$ is the random variable of intermediate features extracted by the adversarially pre-trained teacher model. We employ a Gaussian distribution $\mathcal{N}(\mu_y, \delta)$ with mean $\mu(.)$ and variance $\delta$ as the variational distribution $q(z|t)$. Here, the mean $\mu(.)$ is a function of the intermediate feature from the robust pre-trained model, and the variance $\delta$ is set to an identity matrix as the same CEB. Eq. (7) provides a variational lower bound on IBD, and a more detailed derivation is available in Appendix B.

---

[1]For more details of the bound, we refer the reader to [2], or see Appendix A.

## 3.2 Optimization for IBD

In a standard classification task, we denote $h_\theta = (g \circ f)$ as a deep neural network with parameter $\theta$. Here, $f : \mathbb{R}^{d_X} \to \mathbb{R}^{d_Z}$ maps the inputs $X$ to the intermediate features $Z$, and $g : \mathbb{R}^{d_Z} \to \mathbb{R}^{d_Y}$ further maps the intermediate features $Z$ to the final outputs $Y$, so that $Z = f(X)$ and $g(Z) = h_\theta(X)$. Then, we denote the target student network as $h_s = (g_s \circ f_s)$, and the adversarially pre-trained teacher network as $h_t = (g_t \circ f_t)$. Furthermore, a set of intermediate features from different layers of the student model is denoted as $Z_s = f_s(X) = \{z_s^1, z_s^2, \ldots, z_s^m\}$, and a set of intermediate features from different layers of the teacher model is denoted as $Z_t = f_t(X) = \{z_t^1, z_t^2, \ldots, z_t^n\}$. Here, $n$ and $m$ represent the number of layers of the student and teacher models, respectively. Each feature has its own feature map size and channel dimension, denoted as $z \in \mathbb{R}^{C \times H \times W}$, where $C, H$ and $W$ represent the channel numbers, feature height, and width, respectively.

**1) Maximizing $I(Z, Y)$ via Soft Label Distillation.** To optimize Eq.(7) using the gradient descent algorithm, we simplify the first term on the right-hand side of Eq.(7) to the expected log-likelihood in the form of cross-entropy, which can be formulated as follows:

$$
\begin{aligned}
\mathop{\mathbb{E}}_{p(x,y)p(z|x)} \Big[ \log q(y|z) \Big] &= \int p(y, z) \log q(y|z) dy dz \\
&= \int p(z) p(y|z) \log q(y|z) dy dz \\
&= \int p(z, x) p(y|z) \log q(y|z) dy dz dx \\
&= \int p(x) p(z|x) p(y|z) \log q(y|z) dy dz dx \\
&= \mathop{\mathbb{E}}_{p(x)p(z|x)} \Big[ \int p(y|z) \log q(y|z) dy \Big],
\end{aligned}
\tag{8}
$$

where $p(y|z)$ indicates the true likelihood considered as a target label $y$ corresponding to $z$. The $q(y|z)$ is modeled by the classifier of the student model $h_s$. During the training phase, we can use the output probability of the robust teacher model $h_t$ as the soft label $y_t = h_t(x_{\text{nat}})$ to approximate the distribution of $p(y|z)$, despite knowing the label $y$. According to [74], using robust soft-labels is crucial in enhancing robustness. Thus, we rewrite the first term as:

$$
\mathop{\mathbb{E}}_{p(x,y)p(z|x)} \Big[ \log q(y|z) \Big] = \mathop{\mathbb{E}}_{p(x)p(z|x)} \Big[ \int p(y|z) \log q(y|z) dy \Big] = \mathop{\mathbb{E}}_{p(x)} \Big[ -\mathcal{L}_{CE}\big(h_s(x), y_t\big) \Big]. \tag{9}
$$

**2) Maximizing $-I(Z, X|T)$ via Adaptive Feature Distillation.** To optimize the second term on the right-hand side of Eq.(7), the formulation can be written as:

$$
\begin{aligned}
\mathop{\mathbb{E}}_{p(x)p(z|x)} \Big[ \log \frac{p(z|x)}{q(z|t)} \Big] &= \int p(z, x) \log \frac{p(z|x)}{q(z|t)} dz dx \\
&= \int p(x) p(z|x) \log \frac{p(z|x)}{q(z|t)} dz dx \\
&= \mathop{\mathbb{E}}_{p(x)} \Big[ \int p(z|x) \log \frac{p(z|x)}{q(z|t)} dz \Big].
\end{aligned}
\tag{10}
$$

Thus, we optimize the KL divergence between the feature likelihood $p(z|x)$ and the appropriate feature probability $q(z|t)$. Here, we parameterize Gaussian densities $p(z|x)$ and $q(z|t)$ using neural networks, where the mean of $p(z|x)$ and $q(z|t)$ are the intermediate features of $f_s$ and $f_t$, respectively. Both variances are set to an identity matrix. As a result, the second term can be calculated by:

$$
\mathop{\mathbb{E}}_{p(x,y)p(z|x)} \Big[ \log \frac{p(z|x)}{q(z|t)} \Big] = \mathop{\mathbb{E}}_{p(x)p(z|x)} \Big[ \text{KL}\big(p(z|x) \| q(z|t)\big) \Big] = \mathop{\mathbb{E}}_{p(x)} \Big[ \big(f_t(x) - f_s(x)\big)^2 + c \Big], \tag{11}
$$

where $c$ is a constant. When optimizing DNNs using Eq.(11) as the objective, a challenge arises in selecting suitable intermediate features from models to calculate the loss. This is due to the fact that different intermediate features tend to have different information, especially when the student and teacher models have different architectures. Motivated by [6, 32], we leverage an attention-based feature distillation strategy to achieve cross-layer information transfer.

Given two sets of the intermediate features, $Z_t = f_t(x) = \{z_t^1, z_t^2, \ldots, z_t^n\}$ and $Z_s = f_s(x) = \{z_s^1, z_s^2, \ldots, z_s^m\}$, our aim is to identify similarities for all possible combinations and transfer relevant information from the teacher to the student. We compare the features of both teacher and student models by using two different pooling methods – global average pooling and channel-wise pooling. The similarity determined by two globally pooled features is used as the weight for transferring information over the distance defined by the channel-wisely averaged features. In order to identify the similarity between $Z_t$ and $Z_s$, we adopt a query-key concept of the attention mechanism [61]. In detail, each teacher features generate a query set $Q_t = \{q_t^1, q_t^2, \ldots, q_t^n\}$, and each student feature generate a key set $K_s = \{k_s^1, k_s^2, \ldots, k_s^m\}$, where the $q_t^n$ and $k_s^m$ are calculated as:

$$
\begin{aligned}
\mathbf{q}_t^n &= Re\big(W_t^n \cdot \text{GAP}\,(z_t^n)\big), \\
\mathbf{k}_s^m &= Re\big(W_s^m \cdot \text{GAP}\,(z_s^m)\big),
\end{aligned}
\tag{12}
$$

where GAP denotes the global average pooling, $Re$ is Relu activation function, $W_t^n$ and $W_s^m$ are linear transition parameters. It should be noted that these features possess varying transition weights, as they convey different levels of information.

By utilizing the queries and keys, attention values that represent the relation between teacher and student features are calculated with a softmax function:

$$
Attn = \text{softmax}\left(\frac{Q_T W^{Q-k} K_S^\top}{\sqrt{d}}\right),
\tag{13}
$$

where $W^{Q-k} \in \mathbb{R}^{d \times d}$ is a bilinear weight. The bilinear weighting is utilized to generalize the attention values derived from different source ranks, as queries and keys are identified within features of distinct dimensions [34]. $Attn_{i,j}$ indicates the attention weight that captures the relation between the $i$-th teacher feature and the $j$-th student feature. Therefore, $Attn$ can make the teacher feature $z_t^n$ transmit its corresponding information selectively and adaptively to different student features. Finally, the second term on the right-hand side of Eq. (7) can be written as:

$$
\mathbb{E}_{p(x,y)p(z|x)}\left[\log \frac{p(z|x)}{q(z|t)}\right] = \mathbb{E}_{p(x)}\left[\sum_i^n \sum_j^m Attn_{i,j}\big(\mathbf{T}_t^i(z_t^i) - \mathbf{T}_s^j(z_s^j)\big)^2\right],
\tag{14}
$$

where $\mathbf{T}$ is a transform function with a feature map size alignment (up-sampled or down-sampled), and then apply a channel-wise average pooling to get an average feature map for loss computation.

Combing Eq.9 and Eq.14, the final objective of IBD can be defined as follows:

$$
L_{IBD} = \min_{p(x)} \mathbb{E}\left[\mathcal{L}_{CE}\big(h_s(x), y_t\big) + \beta \sum_i^n \sum_j^m Attn_{i,j}\big(\mathbf{T}_t^i(z_t^i) - \mathbf{T}_s^j(z_s^j)\big)^2\right].
\tag{15}
$$

### 3.3 Applying IBD to Robust Learning

IBD can be naturally applied in combination with adversarial training. The final objective function is formulated as follows:

$$
\begin{aligned}
L_{obj} = \min_{p(x)} \mathbb{E}\Big[&(1-\alpha)\mathcal{L}_{CE}\big(h_s(x_{\text{nat}}), y_t\big) + \alpha\mathcal{L}_{CE}\big(h_s(x_{\text{adv}}), y_t\big) \\
&+ \beta \sum_i^n \sum_j^m Attn_{i,j}\big(\mathbf{T}_t^i(z_t^i) - \mathbf{T}_s^j(z_s^j)\big)^2\Big],
\end{aligned}
\tag{16}
$$

where $\alpha$ and $\beta$ are two trade-off hyper-parameters. We generate adversarial examples using the same method introduced by [71], where given a natural input $x_{\text{nat}}$, generated the adversarial example $x_{\text{adv}}$ by maximizing KL-divergence term. Note that the intermediate features $Z_t$ and $Z_s$ are extracted from $x_{\text{adv}}$. Finally, we utilize this new loss function to train a robust model.

### 3.4 Discussion

The objective of IBD is comparable to certain conventional KD techniques [6, 10, 32]. However, the major difference is, that in our IBD, the teacher model must be an adversarially pre-trained model that can provide robust information. Additionally, we design the objective function following the

Table 1: Robustness comparison of the proposed approach and baseline models under different attack methods under the $\ell_\infty$ norm with $\epsilon = 8/255$ on different datasets. All the models are based on pre-activation ResNet-18 architecture. We choose the best checkpoint according to the highest robust accuracy on the test set under PGD-10. The best results are **blodfaced**.

| Method | Best Checkpoint | | | | | Last Checkpoint | | | | |
|---|---|---|---|---|---|---|---|---|---|---|
| | Clean | FGSM | PGD | CW | AA | Clean | FGSM | PGD | CW | AA |
| **CIFAR-10** - $l_{\inf} - \epsilon = 8/255$ | | | | | | | | | | |
| SAT [41] | 82.97 | 57.77 | 50.85 | 50.09 | 47.73 | 85.16 | 53.97 | 43.03 | 43.71 | 41.58 |
| TRADES [71] | 83.74 | 59.54 | 52.73 | 50.94 | 49.58 | 84.11 | 58.72 | 49.97 | 49.05 | 47.02 |
| CEB [19] | 82.87 | 58.61 | 52.94 | 50.22 | 48.87 | 83.39 | 57.74 | 48.79 | 47.67 | 45.85 |
| HBaR [64] | **84.13** | 58.95 | 53.35 | 51.47 | 49.77 | **85.04** | 58.33 | 53.19 | 51.21 | 48.64 |
| InfoAT [68] | 83.17 | 60.52 | 54.29 | 51.62 | 49.92 | 83.23 | 60.40 | 53.94 | 51.17 | 49.67 |
| ARD [21] | 83.94 | 59.32 | 52.16 | 51.21 | 49.17 | 84.32 | 59.35 | 51.41 | 51.22 | 48.89 |
| IAD [73] | 83.24 | 59.34 | 54.24 | 51.92 | 50.63 | 83.76 | 59.17 | 53.84 | 51.60 | 50.17 |
| RSLAD [74] | 83.38 | 60.08 | 54.27 | 53.19 | 51.52 | 83.88 | 59.98 | 54.01 | 53.08 | 51.36 |
| **IBD** (Ours) | 83.17 | **60.75** | **55.13** | **53.62** | **52.11** | 82.95 | **60.94** | **55.04** | **53.49** | **52.05** |
| **CIFAR-100** - $l_{\inf} - \epsilon = 8/255$ | | | | | | | | | | |
| SAT [41] | 57.75 | 32.78 | 29.27 | 27.52 | 24.01 | 58.17 | 27.02 | 21.02 | 21.54 | 19.86 |
| TRADES [71] | 58.57 | 32.84 | 29.88 | 26.29 | 25.27 | 57.11 | 32.05 | 28.12 | 25.47 | 24.52 |
| CEB [19] | 55.17 | 32.36 | 30.12 | 26.35 | 25.36 | 55.74 | 30.84 | 26.59 | 24.51 | 23.16 |
| HBaR [64] | 59.53 | 34.46 | 31.82 | 27.42 | 26.62 | 58.26 | 32.41 | 28.23 | 25.78 | 25.43 |
| InfoAT [68] | 58.23 | 34.46 | 31.39 | 28.68 | 26.76 | 58.42 | 33.13 | 30.53 | 27.52 | 26.21 |
| ARD [21] | **60.58** | 33.43 | 29.07 | 27.54 | 25.62 | **60.79** | 32.67 | 28.11 | 26.76 | 24.62 |
| IAD [73] | 57.08 | 34.65 | 30.60 | 27.24 | 25.84 | 57.52 | 33.71 | 29.23 | 27.35 | 25.36 |
| RSLAD [74] | 57.72 | 34.23 | 31.01 | 28.27 | 26.73 | 57.83 | 34.09 | 30.55 | 28.07 | 26.41 |
| **IBD** (Ours) | 58.10 | **36.37** | **33.59** | **31.16** | **29.21** | 58.32 | **36.17** | **33.40** | **30.87** | **28.74** |

information bottleneck principle, which is theoretically proven to be a lower bound of the information bottleneck. Notably, previous adversarial distillation methods [21, 44, 74] only consider utilizing information from the final prediction output, while ignoring the information from intermediate features. From another perspective, motivated by [31], IBD can explore intermediate features to capture the robust information and leverage an adaptive feature distillation strategy to automatically transfer appropriate features from the teacher model to the target student model. Our experiments indicate that making full use of the intermediate features of the DNNs can improve the model's robustness effectively.

## 4 Experiments

In this section, we first describe the experimental setting and implementation details. We evaluate the robustness and accuracy of various widely used benchmark datasets. Finally, we conduct a wealth of ablation experiments to provide a comprehensive understanding of the proposed IBD.

### 4.1 Experiments Settings

**Baselines Setup.** We conduct our experiments on three benchmark datasets including CIFAR-10, CIFAR-100 [36] and ImageNet [17]. We consider three types of baseline model: 1) classical adversarial training method, including standard Standard AT[41] and TRADES [71]; 2) adversarial robustness distillation, including ARD [21], IAD [73] and RSLAD [74]; 3) the models trained using variants IB, include CEB [19], HBaR [64] and InfoAT [68].

**Evaluation Attack.** We evaluate the defense under different white- and black-box attacks including FGSM [22], PGD [41], CW [4], and AutoAttack [14]. The black-box attacks include query-based

Table 2: The performance of the pre-trained teacher models under different attacks.

| Dataset | Architecture | Natural Acc | FGSM | PGD-100 | CW-100 | AutoAttack |
|---------|-------------|-------------|------|---------|--------|------------|
| CIFAR-10 | WideResNet-34-10 | 84.92 | 60.87 | 55.33 | 53.98 | 53.08 |
| CIFAR-100 | WideResNet-34-10 | 57.16 | 33.58 | 30.61 | 27.74 | 26.78 |
| ImageNet-1K | ResNet-50 | 64.02 | - | 38.46 | - | 34.96 |

Table 3: Black-box robustness results in CIFAR-10 under the $\ell_\infty$ norm with $\epsilon = 8/255$. We evaluate against transfer-based and query-based attacks.

| Method | Trans-based | | Query-based | |
|--------|------|---------|------|--------|
| | FGSM | PGD-100 | SPSA | Square |
| SAT[41] | 61.34 | 59.83 | 66.35 | 54.16 |
| TRADES[71] | 63.14 | 60.31 | 67.56 | 54.74 |
| CEB[19] | 62.75 | 61.93 | 66.71 | 55.28 |
| HBaR[64] | 63.61 | 60.93 | 68.07 | 56.62 |
| InfoAT [68] | 64.23 | 62.42 | 68.73 | 57.35 |
| ARD [21] | 63.27 | 61.44 | 67.89 | 55.85 |
| IAD [73] | 63.53 | 62.13 | 68.42 | 56.21 |
| RSLAD[74] | 64.78 | 62.13 | 68.78 | 56.97 |
| **IBD** | **65.54** | **63.98** | **69.63** | **58.34** |

Table 4: Robustness comparison of the proposed IBD and several state-of-the-art models under standard AutoAttack.

| Method | WRN | Clean | AA |
|--------|-----|-------|-----|
| SAT [41] | 34-10 | 84.92 | 53.08 |
| LBGAT [15] | 34-20 | 88.70 | 53.57 |
| TRADES [43] | 34-20 | 86.18 | 54.39 |
| LTD [7] | 34-10 | 85.02 | 54.45 |
| **IBD** | 34-10 | 83.33 | **55.65** |
| TRADES + AWP [66] | 34-10 | 85.26 | 56.17 |
| LASAT + AWP [33] | 34-10 | 84.98 | 56.26 |
| LTD + AWP [7] | 34-10 | 86.28 | 56.94 |
| **IBD** + AWP | 34-10 | 85.21 | **57.18** |

attacks [1] and transfer-based attacks [59]. The maximum pertubation is set to $\epsilon = 8/255$ for CIFAR-10 and CIFAR-100.

**Implementation Details.** For the robust pre-trained teacher models on the CIFAR dataset, we use the ResNet-18 [26] and WideResNet-34x10 [70] models trained with a way of TRADES [71] and AWP [66]. For the ImageNet dataset, the pre-trained teacher model (a ResNet-50) is provided by [49]. The results of the pre-trained teacher models under different attacks are shown in Table 8. The initial learning rate is 0.1 with a piece-wise schedule which is divided by 10 at epochs 100 and 150 for a total number of 200 training epochs, similar to [47]. We train all models with the SGD optimizer with a momentum of 0.9, weight decay of 0.0005, and a batch size of 128. We use ResNet-18 as the student model for most experiments by default, unless otherwise stated. We adopt the common setting that the $\ell_\infty$ threat model with radius $8/255$, with the PGD attack taking 10 steps of size $2/255$. In addition, we performed standard data augmentation, including random crops and random horizontal flips during training. For the hyper-parameter, we set $\alpha = 0.9$ and $\beta = 0.8$ based on our ablation studies. For more details please refer to our open source code. Our implementation is based on PyTorch and the code to reproduce our results is available at https://github.com/SkyKuang/IBD.

## 4.2 Adversarial Robustness Evaluation

**White-box Robustness.** To verify the impact of the IBD on model robustness, we first train a natural model (without adversarial training), We evaluate the performance of IBD on CIFAR-10 under ResNet-18, and IBD achieves $25.49\%$ robust accuracy against standard AA. However, other IB-based methods (CEB and HBaR) without adversarial training can not defend AA (that means the robust accuracy is 0). We further combine IBD with adversarial training and evaluate the robustness of all baseline models and our IBD against various types of attacks. Following in [47], we report the results at both the best checkpoint and the last checkpoint. The best checkpoint is selected based on the performance under the PGD-10 attack. The results are shown in Table 1. Our IBD method shows better robustness of both CIFAR-10 and CIFAR-100 against all attacks at either the best or the last checkpoints. In particular, our IBD improves the robustness by $0.59\%$ and $2.48\%$ on CIFAR-10 and CIFAR-100 respectively, compared to previous state-of-the-art methods under AutoAttack.

**Black-box Robustness.** To further verify the robustness of our method under black-box attacks, we evaluate our IBD and baseline methods against transfer-based attacks and query-based attacks. For transfer-based attacks, we choose a robust surrogate model, which is trained by standard adversarial training, to generate the adversarial examples. All the models are trained using the same setting

(referring to Section 4.1). We generate adversarial examples using both FGSM and PGD-100 attacks with attack budget $\epsilon = 8/255$ on CIFAR-10. For query-based attacks, we evaluate the robustness under SPSA attack [60] and Square attack [13]. SPSA attack can make a full gradient evaluation by drawing random samples and obtaining the corresponding loss values. We set the number of random samples $q$ as 128 for every iteration and $\delta = 0.01$ and set iteration as 10. The square attack uses random search and does not exploit any gradient approximation to produce adversarial noise, and we set the maximum number of queries as 5000. We evaluate both transfer-based and query-based attacks on the best checkpoints. The results are shown in Table 3. We observe that the IBD-trained model is robust to different types of black-box attacks and surpasses all baseline methods. The results under the black-box attack demonstrate the robustness improvement of IBD is not caused by the obfuscated gradients.

**Robustness Evaluation on WideResNet.** Many works have demonstrated larger model capacity can usually lead to better adversarial robustness [23, 41, 43]. Therefore, we employ the large-capacity network, *e.g.*, WideResNet (WRN) [70], as the student model. Table 4 reports the robustness results against AA on the CIFAR-10. We compare several state-of-the-art adversarial trained models on robust benchmark [12]. We can observe that the proposed IBD indeed improves the adversarial robustness by ~ 1.2%. Furthermore, when combined with AWP [66], our IBD also surpasses the previously state-of-the-art models reported by the benchmark. where every small margin of improvement is significant. **Note that**, our method does not use any additional datasets.

**Evaluation on Large-scale ImageNet.** To further verify the generalization of our method, we consider a high-resolution, large-scale ImageNet dataset[17], which includes 1,000 classes and more than 1M training samples. Maintaining robustness for this dataset is particularly challenging. We use the fast adversarial training framework [65] to train all robust models. Here, the robust budgets are set as $\epsilon = 2/255$ and $\epsilon = 4/255$. The results of our method for the ResNet-50 (student model) are shown in Table 5. Our IBD outperforms both Fast-AT [62] and RSLAD [74], significantly.

Table 5: Robustness comparison of the proposed approach and different fast adversarial training under different attack methods at $\epsilon = 2/255$ and $\epsilon = 4/255$ on ImageNet-1k. All the models are based on ResNet-50 architecture.

| Method | Epsilon | Clean | PGD-100 | AutoAttack |
|---|---|---|---|---|
| FAST-AT [65] | $\epsilon = 2$ | **65.95** | 37.52 | 35.22 |
| RSLAD [74] | $\epsilon = 2$ | 63.64 | 40.49 | 38.35 |
| FAST-IBD | $\epsilon = 2$ | 62.03 | **44.31** | **40.94** |
| FAST-AT [65] | $\epsilon = 4$ | 60.16 | 27.46 | 24.82 |
| RSLAD [74] | $\epsilon = 4$ | **60.64** | 30.19 | 26.53 |
| FAST-IBD | $\epsilon = 4$ | 59.10 | **31.52** | **27.74** |

## 4.3 Ablation Studies

**The importance of robust soft label.** [52] proposes leveraging label smoothing during adversarial training, and [74] also determines the robust soft label is an important factor in robustness enhancement. We empirically verify the importance of robust soft labels in our IBD, by comparing the performance of models trained using different label modification schemes: a) true label (hard label); b) smoothing label crafted by label smoothing on the true label [55]; c) natural soft label that output by non-robust pre-trained model; 4) robust soft label that output by robust pre-trained model. The results are summarized in Table 6, which demonstrates that the robust soft label is beneficial to improving the model's robustness.

**The role of features matching.** We further investigate the role of features distillation strategy in section 3.2, we consider two features matching strategies: a) a hand-crafted feature distillation method, which uses the same level of features to guide the student model learning; b) an adaptive feature distillation method, which is based on attention mechanism to guide the student model learning. We conduct experiments with these two strategies on different network architectures ResNet-18 (Res) and WideResNet-34x10 (WRN). The results are reported in Table 7. Our adaptive feature

Table 6: Robustness of our IBD model trained using different types of labels on CIFAR-10 under different attacks.

| Method | Clean | PGD | CW | AA |
|---|---|---|---|---|
| Hard label | 85.25 | 51.64 | 50.49 | 48.35 |
| Label smooth | 85.43 | 51.92 | 50.56 | 48.71 |
| Natural soft label | 86.38 | 48.08 | 45.70 | 43.79 |
| Robust soft label | 83.17 | 55.13 | 53.62 | 52.11 |
| RSLAD [74] | 83.38 | 54.27 | 53.19 | 51.52 |

Table 7: Robustness of our IBD model trained using different features matching schedule on CIFAR-10 under the AutoAttack.

| Architecture | Schedule | Clean | AA |
|---|---|---|---|
| Res → Res | No matching | 83.25 | 50.63 |
| WRN → Res | No matching | 83.37 | 51.48 |
| Res → Res | Hand-crafted | 82.99 | 51.12 |
| WRN → Res | Hand-crafted | 83.10 | 51.61 |
| Res → Res | Adaptive | 83.24 | 51.46 |
| WRN → Res | Adaptive | 83.17 | 52.11 |

distillation performs better than hand-crafted feature distillation in both the same and different network architectures.

**The impact of the teacher.** We conducted an ablation experiment by using different teacher models to verify the impact of the teacher's robustness on the performance of the student model. We conduct this experiment on CIFAR-10 with two student models: ResNet-18 and WideResNet-34-10, and five different teacher models which have different robustness. The results are shown in Table 8 in the Appendix. We can observe that different robust teacher models have a significant positive benefit on the student model. For the ResNet-18 student model, we find that the robustness of the student does not increase monotonically with that of the teacher. As the teacher model (WideResNet-34-20) becomes more complex, the robustness of the student model decreases, compared to WideResNet-34-10. This may be due to the large gap in the architecture of the teacher model and the student model. This phenomenon is called robust saturation [74]. For the WideResNet-34-10 student model, we found that in most cases, the student's robustness can surpass that of the teacher model. We think there are two reasons for this, one is that the performance of the teacher model is not very strong. The other is that the teacher model provides robust soft labels to alleviate overfitting and improve performance. Therefore, in most cases, it is expected that the student model exceeds the teacher model, but when the teacher model is strong enough, it is not easy for the student model to surpass the teacher model (e.g., WideResNet-76-10).

**The impact of $\alpha$ and $\beta$.** The $\alpha$ is a trade-off the adversarial robustness and natural accuracy. We conduct ablation experiments to verify the trade-off. The results are shown in Figure 2(a) in the Appendix. When we set $\alpha = 0.9$, our method can achieve the best adversarial robustness. In the IB principle, the hyperparameters $\beta$ are important to control the trade-off. Therefore, we finally choose different values of $\beta$ to train the IBD, from which we choose the optimal value of $\beta$. The experimental results are shown in Figure 2(b) in the Appendix, where the best results are achieved when setting $\beta = 0.8$.

## 5   Conclusions

In this paper, we revisited the information bottleneck principle from the perspective of robustness distillation and then presented a new IB objective, called Information Bottleneck Distillation (IBD). IBD can be thought of as a tighter variational approximation to the IB objective than VIB. To optimize IBD effectively, we proposed two adaptive distillation strategies. Experimentally, we empirically demonstrated the advantage of IBD over existing methods, IB-based methods, and adversarial distillation methods on benchmark datasets.

## 6   Acknowledgments

This work was supported by National Key R&D Program of China (No.2022ZD0118202), the National Science Fund for Distinguished Young Scholars (No.62025603), the National Natural Science Foundation of China (No. U21B2037, No. U22B2051, No. 62176222, No. 62176223, No. 62176226, No. 62072386, No. 62072387, No. 62072389, No. 62002305 and No. 62272401), and the Natural Science Foundation of Fujian Province of China (No.2021J01002, No.2022J06001).

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

## A  Variational Information Bottleneck

We show the derivation process of Variational Information Bottleneck (VIB), which is provided by [2, 35]. The information bottleneck (IB) [57] expresses a trade-off in intermediate features $Z$ between information useful for output prediction $Y$ and information retained about the input $X$. The objective of IB can be formulated as follows:

$$IB(\theta) = \max I(Z;Y) - \beta I(Z;X), \tag{17}$$

where $I$ denotes mutual information and $\beta$ controls the trade-off between the two terms.

The first term can be expressed as follows:

$$
\begin{aligned}
I(Z;Y) &= \int p(y,z) \log \frac{p(y,z)}{p(y)p(z)} dy dz \\
&= \int p(y,z) \log \frac{p(y|z)}{p(y)} dy dz
\end{aligned}
\tag{18}
$$

To apply it to a deep neural network $h_\theta = (g * f)$, The $q(y|z)$ is modelled by the classifier $g$. Based on variational inference, it is a closed form for a true likelihood $p(y|z)$. This approximation is formulated by KL divergence and then the following inequality is constructed as follows:

$$KL\big(p(Y|Z)\|q(Y|Z)\big) \geq 0 \quad \Rightarrow \quad \int p(y|z)\log p(y|z) dy \geq \int p(y|z)\log q(y|z) dy \tag{19}$$

which helps to get the objective of IB to be tractable. With this equality, the first term in Eq. (17) can be represented to a lower bound as:

$$
\begin{aligned}
I(Z;Y) &\geq \int p(y,z) \log \frac{q(y|z)}{p(y)} dy dz \\
&= \int p(y,z) \log q(y|z) dy dz - \int p(y,z) \log p(y) dy dz \\
&= \int p(y,z) \log q(y|z) dy dz - \int p(y) \log p(y) dy \\
&= \int p(y,z) \log q(y|z) dy dz + H(Y) \\
&\geq \int p(y,z) \log q(y|z) dy dz = \mathbb{E}_{p(x,y)p(z|x)}\big[\log q(y|z)\big]
\end{aligned}
\tag{20}
$$

where a positive constant $H(Y)$ denotes the Shannon entropy of target labels.

The second term in Eq. (17) is described as :

$$
\begin{aligned}
I(Z;X) &= \int p(z,x) \log \frac{p(z,x)}{p(x)p(z)} dz dx \\
&= \int p(z,x) \log \frac{p(z \mid x)}{p(z)} dz dx
\end{aligned}
\tag{21}
$$

where a dataset probability $p(x)$ is erased on the fraction. Here, an approximate feature probability $q(Z)$ is introduced to appropriate the true feature probability $p(Z)$. As similar to Eq.(19), the relationship between $q(Z)$ and $p(Z)$ can be written and then it builds the following equality:

$$KL\big(p(Z)\|q(Z)\big) \geq 0 \quad \Rightarrow \quad \int p(z) \log p(z) dz \geq \int p(z) \log q(z) dz \tag{22}$$

By using it, the second term is constructed with an upper bound as follows:

$$
\begin{aligned}
I(Z;X) &\leq \int p(z,x) \log \frac{p(z \mid x)}{q(z)} dz dx \\
&= \int p(x)p(z \mid x) \log \frac{p(z \mid x)}{q(z)} dz dx \\
&= \mathbb{E}_{p(x,y)p(z|x)}\left[\log \frac{p(z|x)}{q(z)}\right]
\end{aligned}
\tag{23}
$$

where a feature likelihood is denoted by $p(z|x)$. To sum it up, the objective of IB can be re-formulated with a lower bound as follows:

$$I(Z;Y) - \beta I(Z;X) \geq \mathbb{E}_{p(x,y)p(z|x)}\left[\log q(y|z) - \beta \log \frac{p(z|x)}{q(z)}\right], \tag{24}$$

## B  Variational Bound on IBD

In the IBD setting, there is a target model and a pre-trained teacher model. We would like to learn features $Z$ of X that will be useful for predicting $Y$ for the target model. $T$ is the intermediate features extracted by the adversarially pre-trained teacher model. We can represent this problem setting with the Markov chain: $Z \leftarrow X \rightarrow T$. Given the conditional independence $Z \perp\!\!\!\perp T|X$ in our Markov chain and the conditional mutual information is always non-negative [20], so learning intermediate features $Z$ of $X$ is equivalent to minimizing $I(X;Z|T)$. Using the chain rule of mutual information, we have:

$$I(X;Z|T) = I(X,T;Z) - I(T;Z) = I(X;Z) - I(T;Z) \tag{25}$$

The objective of IBD can be formulated as:

$$IBD = I(Z;Y) - \beta I(Z;X|T) \tag{26}$$
$$= H(Y) + H(Y|Z) - \beta\big(H(Z) - H(Z|X) - H(Z) + H(Z|T)\big) \tag{27}$$
$$= H(Y|Z) - \beta\big(H(Z|T) - H(Z|X)\big) \tag{28}$$

We will variationally lower bound the first term of Eq. (26) and upper bound the second term using three distributions: $p(z|x)$ is the feature distribution; $q(z|t)$ is an approximation of $p(z|y)$; and $q(y|z)$ is an approximation of $p(y|z)$.

The first term of Eq. (26):

$$
\begin{aligned}
I(Z;Y) &= H(Y) - H(Y|Z) \\
&= \mathop{\mathbb{E}}_{p(x,y)p(z|x)}\Big[\log q(y|z) + KL\big(p(y|z)\|q(y|z)\big)\Big] \\
&\geq \mathop{\mathbb{E}}_{p(x,y)p(z|x)}\Big[\log q(y|z)\Big].
\end{aligned}
\tag{29}
$$

The second term of Eq. (26) (for convenience, we set $\beta$ to 1):

$$
\begin{aligned}
I(X;Z|T) &= H(Z|T) - H(Z|X) \\
&= \mathop{\mathbb{E}}_{p(x)p(z|x)}\Big[\log p(z|x) - \log q(z|t) - KL\big(p(z|y)\|q(z|t)\big)\Big] \\
&\leq \mathop{\mathbb{E}}_{p(x)p(z|x)}\Big[\log \frac{p(z|x)}{q(z|t)}\Big].
\end{aligned}
\tag{30}
$$

So far, we get the variational bound for IBD:

$$IBD = I(Z;Y) - \beta I(X;Z|T) \geq \mathop{\mathbb{E}}_{p(x,y)p(z|x)}\Big[\log q(y|z) - \beta \log \frac{p(z|x)}{q(z|t)}\Big], \tag{31}$$

## C  More Experimental Details and Results

### C.1  The impact of the teacher.

Table 8:  How would the performance of teachers affect that of student models?

| Teacher | Natural | AA | Student | Natural | AA | Student | Natural | AA |
|---|---|---|---|---|---|---|---|---|
| Resnet-18 | 84.09 | 48.71 | Resnet-18 | 83.74 | 50.52 | WRN-34-10 | 84.41 | 53.94 |
| Resnet-34 | 85.94 | 50.57 | Resnet-18 | 84.92 | 49.84 | WRN-34-10 | 85.79 | 54.17 |
| WRN-34-10 | 84.92 | 53.08 | Resnet-18 | 83.17 | 52.11 | WRN-34-10 | 84.21 | 55.65 |
| WRN-34-20 | 85.65 | 56.82 | Resnet-18 | 82.82 | 51.64 | WRN-34-10 | 84.73 | 55.71 |
| WRN-76-10 | 88.54 | 64.25 | Resnet-18 | 85.28 | 51.96 | WRN-34-10 | 86.61 | 57.12 |

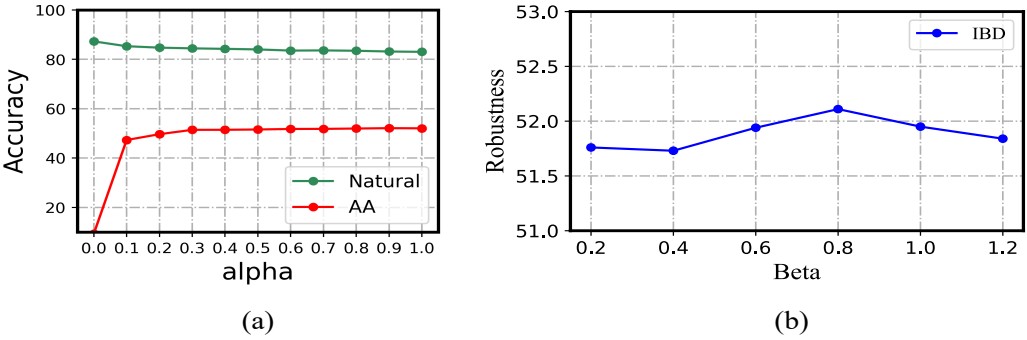

Figure 2: The impact of $\alpha$ and $\beta$.

## C.2 The impact of $\alpha$ and $\beta$.

## C.3 Different attack budgets and steps

We plot the results of testing robust accuracy over epochs and evaluate adversarial accuracy against PGD attacks under different attack budgets with a fixed attack step of 10, and we also conduct experiments using PGD attacks with different attack iterations with a fixed attack budget of 8/255. The results are shown in Figure 3. Our IBD is better than standard AT, TRADES and RSLAD at larger budgets, besides, our IBD is stable against large iterations attacks, *e.g.,* PGD attack with 500 step iterations. Therefore, the results demonstrate the effectiveness of our proposed IBD.

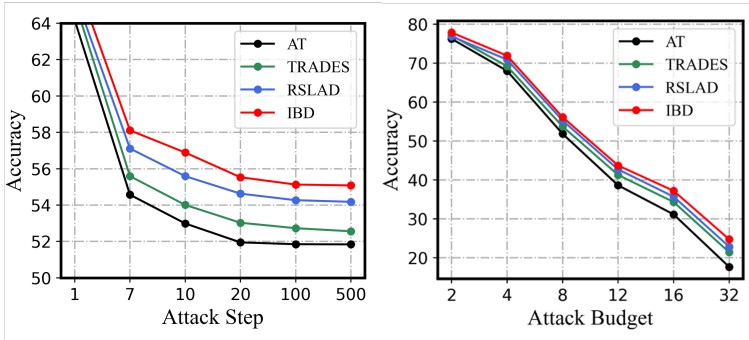

Figure 3: The test robust accuracy under PGD attack with different attack budgets and attack iterations, respectively. All these experiments were conducted on CIFAR-10 using ResNet-18 architecture. (Best view in color)

## C.4 Visualization of attention matrix

We visualize the attention matrices when distilling between different network architectures, as shown in Figure 4. We can see that models of the same architecture can achieve mutual correspondence between feature layers, but for models of different architectures, the shallow layers of the student model will receive a lot of attention, and the high-level features will correspond to each other.

# D Limitations.

One limitation of our method is that the accuracy of our method on natural samples does not improve significantly, This may be because there exists a trade-off between robustness and natural accuracy [71]. Despite this, our natural accuracy is still over 83+% on CIFAR10, we think this is acceptable. Another limitation is that our method requires a robust pre-trained teacher model, which may increase training costs and training time. At present, our IBD has only verified its effect on the adversarial

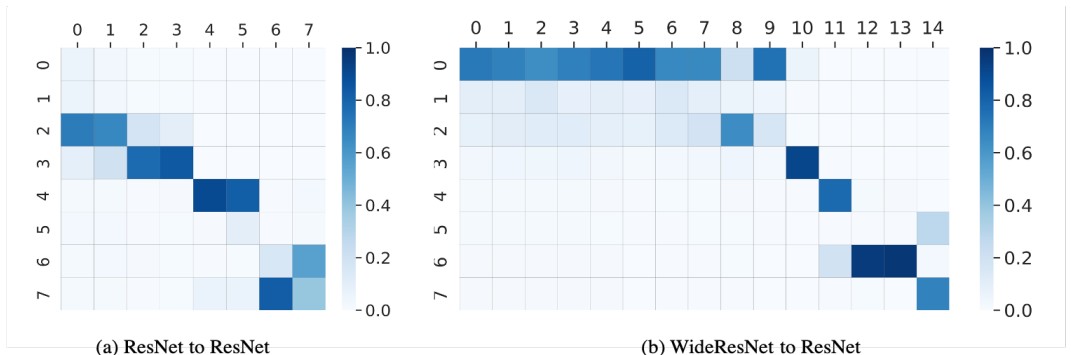

|             | (a) ResNet to ResNet | (b) WideResNet to ResNet |
| ----------- | -------------------- | ------------------------ |

Figure 4: The visualization of attention matrix.

robustness of DNNs. It is not yet known whether it can achieve the same effect in other applications of IB. We are further studying its generalization ability.

## E    Broader Impacts.

We propose an adversarial defense method to enhance the robustness of the model, but we still need to be aware of the potential negative societal impacts it might result in. For example, the attacker can get our model and design a special attack algorithm for it. At present, we can not guarantee that our model can defend against stronger attack algorithms that may appear in the future. Thus, we encourage our machine learning community to further establish more reliable adversarial robustness checking routines for machine learning models deployed in safety-critical applications.

