# OpenReview forum: "Improving Adversarial Robustness via Information Bottleneck Distillation"
_NeurIPS.cc/2023/Conference — NeurIPS 2023 poster_

### Official Review · Reviewer_Dvz1 · 2023-07-05

**Soundness:** 3 good
**Presentation:** 3 good
**Contribution:** 3 good
**Rating:** 6
**Confidence:** 4

**Summary:**

This paper proposes Information Bottleneck Distillation (IBD) to improve adversarial robustness from the perspective of information bottleneck principle. Specifically, two distillation strategies are proposed to boost information bottleneck. Different from the existing works, this paper utilizes the predictions of robust models to maximize the mutual information. Also, the authors design an adaptive feature distillation based on the attention mechanism to facilitate the student model to inherit knowledge from the teacher model. The experimental results demonstrate the proposed strategies are effective in improving adversarial robustness.

**Strengths:**

S1: This paper is technically sound, and the motivation and formulation of the proposed method are elegant.

S2: Formulating the information bottleneck objective from the perspective of adversarial robustness distillation is novel and well-motivated.

S3: The difference with the existing works is clearly demonstrated and discussed.

**Weaknesses:**

W1: Some derivation details in this paper are overly simplified and jumpy, making it difficult to understand. For example, how is the last step of the derivation of Eq. (11) obtained?

W2: The proposed methods rely heavily on a robust teacher model, hence the experimental evaluation of what effect of varying teachers on the proposed methods should be conducted.


**Questions:**

As mentioned in W1 and W2, how is the last step of the derivation of Eq. (11) obtained? And, what is the effect of varying teachers on the proposed methods ?

**Limitations:**

The derivation details are not very clear.

---

> ### Author Rebuttal · Authors · 2023-08-10
>
> Thanks for your valuable feedback! Here is our response to the concerned questions.
> #### **Q1: Some derivation details in this paper are overly simplified and jumpy**
> **A1**:Thank you for your comment and sorry for the confusion. To clear up this, we further provide the detailed derivation of Eq.(11). The Eq. (11) is
> \begin{equation}
>     \begin{aligned}
>   \mathop{\mathbb{E}}\limits_{p(x)p(z|x)}  \left[ \log \frac{p(z | x)}{q(z | t)}\right] = \mathop{\mathbb{E}}\limits_{p(x)}  \left[ \int p(z | x) \log \frac{p(z | x)}{q(z|t)} d z  \right] =
>   \mathop{\mathbb{E}}\limits_{p(x)p(z|x)} \Big[\mathrm{KL} \big(p(z | x) || q(z|t) \big) \Big].
>     \end{aligned}
> \end{equation}
>
> Thus, we optimize the $\mathrm{KL}$ divergence between the feature likelihood $p(z|x)$ and the appropriate feature probability $q(z|t)$.
> In IBD, we parameterize Gaussian densities $p(z|x)$ and $q(z|t)$ using neural networks, where the mean of $p(z|x)$ and $q(z|t)$ are the intermediate features from $f_s$ and $f_t$, respectively, and the variance is set to an identity matrix.
>
> The $\mathrm{KL}$ divergence between two Gaussian distributions $p$ and $q$ can be obtained by
> \begin{equation}
> \begin{aligned}
> \mathrm{KL}(p, q) & =-\int p(x) \log q(x) d x+\int p(x) \log p(x) d x \\
> & =\frac{1}{2} \log \left(2 \pi \sigma_2^2\right)+\frac{\sigma_1^2+\left(\mu_1-\mu_2\right)^2}{2 \sigma_2^2}-\frac{1}{2}\left(1+\log 2 \pi \sigma_1^2\right) \\
> & =\log \frac{\sigma_2}{\sigma_1}+\frac{\sigma_1^2+\left(\mu_1-\mu_2\right)^2}{2 \sigma_2^2}-\frac{1}{2}
> % = \frac{1}{2c_1} \left(\mu_1-\mu_2\right)^2 + c_2,
> \end{aligned}
> \end{equation}
>
> In IBD, $\mu_1$ is $f_t(x)$ and $\mu_2$ is $f_s(x)$. $\sigma_1$ and $\sigma_2$ are identity matrices.
> As a result, the Eq.(11) can be calculated by:
> \begin{equation}
>     \begin{aligned}
>   \mathop{\mathbb{E}}\limits_{p(x,y)p(z|x)}  \left[ \log \frac{p(z | x)}{q(z | t)}\right] = \mathop{\mathbb{E}}\limits_{p(x)p(z|x)} \Big[\mathrm{KL} \big(p(z | x) || q(z|t) \big) \Big]
>    = \mathop{\mathbb{E}}\limits_{p(x)}  \left[  \big(f_t(x) - f_s(x)\big)^2 + \text{c}  \right],
>     \end{aligned}
> \end{equation}
> where $c$ is a constant.
>
> When applying Eq.(11) as an objective to optimize the DNNs, a particular challenge is how to choose appropriate intermediate features from models to calculate the loss, since the different intermediate features tend to have different levels of information, especially when the student and teacher models have different architectures. Therefore, we proposed an attention-based feature distillation strategy to achieve this optimization objective.
>
> **In order to avoid confusion about the derivation process, we will give the detailed derivation steps of each formula in the new version.**
>
> #### **Q2: How would the performance of teachers affect that of student models?**
> **A2**: Thank you for your good questions. In our submitted manuscript, we only used one robust teacher network for fair experimental comparison. Following your comments, we conduct an ablation experiment by using different teacher models to verify the impact of the teacher's robustness on the performance of the student model. We conduct this experiment on CIFAR-10 with two student models: ResNet-18 and WideResNet-34-10, and five different teacher models which have different robustness. The results are shown in the Table. We can observe that different robust teacher models have a significant positive benefit on the student model. For the ResNet-18 student model, we find that the robustness of the student does not increase monotonically with that of the teacher.
> As the teacher model (WideResNet-34-20) becomes more complex, the robustness of the student model decreases, compared to WideResNet-34-10.  This may be due to the large gap in the architecture of the teacher model and the student model.
> This phenomenon is called **Robust saturation** [1].
> For the WideResNet-34-10 student model, we found that in most cases, the student’s robustness can surpass that of the teacher model.
> We think there are two reasons for this, one is that the performance of the teacher model is not very strong.
> The other is that the teacher model provides robust soft labels to alleviate overfitting and improve performance.
> Therefore, in most cases, it is expected that the student model exceeds the teacher model, but when the teacher model is strong enough, it is not easy for the student model to surpass the teacher model (e.g., WideResNet-76-10).
>
>  Teacher   | Natural  | AutoAtt   | Student  | Natural  | AutoAtt    | Student  | Natural   | AutoAtt   |
>  ----  | ----  | ----  | ----  | ----  | ----  | ----  | ----  | ----  |
> Resnet-18  | 84.09 | 48.71 | Resnet-18  | 83.74 | 50.52 | WRN-34-10  | 84.41 | 53.94 |
> Resnet-34  | 85.94 | 50.57 | Resnet-18  | 84.92 | 49.84 | WRN-34-10  | 85.79 | 54.17 |
> WRN-34-10  | 84.92 | 53.08 | Resnet-18  | 83.17 | 52.11 | WRN-34-10  | 84.21 | 55.65 |
> WRN-34-20  | 85.65 | 56.82 | Resnet-18  | 82.82 | 51.64 | WRN-34-10 | 84.73 | 55.71 |
> WRN-76-10  | 88.54 | 64.25 | Resnet-18  | 85.28 | 51.98 | WRN-34-10  | 86.61 | 57.12 |
>
> Ref:
>
> [1]  Revisiting adversarial robustness distillation: Robust soft labels make student better. ICCV 2021.

---

> > ### Comment · Reviewer_Dvz1 · 2023-08-18
> >
> > I appreciate the authors' detailed response to the initial review. Having carefully considered their feedback in conjunction with the comments from other reviewers, I decide to maintain my initial rating.

---

> > > ### Author Response · Authors · 2023-08-19
> > > **Response to Reviewer Dvz1**
> > >
> > > Thank you for considering our response and providing feedback. We greatly appreciate your valuable comments to further strengthen our work. Thank you again for your time and input.

---

### Official Review · Reviewer_wRTW · 2023-07-05

**Soundness:** 2 fair
**Presentation:** 3 good
**Contribution:** 3 good
**Rating:** 5
**Confidence:** 3

**Summary:**

This paper takes a closer look at the information bottleneck principle and show that specially designed robust distillation can boost information bottleneck, benefiting from the prior knowledge of a robust pre-trained model and presents the Information Bottleneck Distillation (IBD) approach. What’s more, this paper also propose two distillation strategies to match the two optimization processes of the IB, respectively. The experimental results demonstrate the effectiveness of IBD.


**Strengths:**

This article is the first to design a distillation objective function for information bottlenecks called IBD objective, and then propose two distillation strategies tor perform the two optimization processes of the information bottleneck for the further use in adversarial training.This article is the first to design a distillation objective function for information bottlenecks



**Weaknesses:**

See Questions part.


**Questions:**

1.The author should explicitly list the main contributions of this work, which would helps readers especially not with prior knowledge of relevant methods quickly understand the core content of the article and the value of the work, and also save reading time.

2.This work proposes IBD by combining robustness distillation and information bottleneck. I am concerned about the motivation of IBD. What problem does IBD solve for robust distillation or information bottleneck in adversarial training tasks?

3.The proposed IBD followed with two distillation strategies corresponding to the two optimization processes of the information bottleneck. Could the author explain more about the these two processes? My understanding is that these two strategies are designed for the two terms of the IB objective which require different optimization methods, and then integrated into the final objective function.

4.From the experimental results of CIFAR-10 in Table 1, although the proposed method has improved the robustness, it seems to be obtained by sacrificing the clean accuracy. Does the author have any ideas for improvement on this trade-off?

---

> ### Author Rebuttal · Authors · 2023-08-10
>
> Thanks for your valuable feedback! Here is our response to the concerned questions.
>
> #### **Q1: The main contributions of our work.**
> **A1**: Thank you for your comment and fruitful advice. The main contributions of our work include as follows:
> 1) **Theoretically**, we utilize conditional variational inference to construct a lower bound to estimate the mutual information and reformat the IB principle by using the adversarial robustness as the prior for learning features, which is termed Information Bottleneck Distillation (IBD).
> 2) **Algorithmically**, to realize IBD, we propose two distillation strategies to match the two optimization processes of the information bottleneck, respectively.
> First, we utilize robust soft label distillation to maximize the mutual information between latent features and output prediction.
> Second, we present an adaptive feature distillation that automatically transfers relevant knowledge from the teacher model to the student
> model, so that it can restrict the mutual information between the input and latent features.
> 3) **Experimentally**, we conducted extensive experiments on various benchmark datasets such as CIFAR and ImageNet.
> The results show the effectiveness of our IBD in improving the robustness of DNNs against most attacks (e.g., PGD-attack and AutoAttack), and our IBD behaves more robustly than state-of-the-art methods.
>
> #### **Q2: What problem does IBD solve for robust distillation or information bottleneck in adversarial training tasks?**
> **A2**: Thank you for your comments and sorry for the confusion.  Our motivation is to improve the robustness of the model through the efficient optimization of IB.
>
> 1) How does IBD improve the optimization of IB? IBD assists the calculation of mutual information $I(x; z)$ by introducing an adversarial robust prior for each sample. Therefore, when calculating mutual information, IBD provides customized and reliable prior information according to different samples, rather a uniform priors. Our purpose is to let the model learn more relevant information and discard nuisance information.
>
> 2) From the perspective of robust distillation, previous adversarial robust distillation methods only optimize the output distribution of the model, while ignoring the feature information of the model.
> Previous work [1] has shown that there are two kinds of features in the neural network: robust and non-robust features.
> Therefore, we adopt the robust model as the teacher model and use the adversarial robustness as the prior for learning robust features.
> We propose an adaptive feature distillation strategy to automatically transfer robust features from the teacher model to the student model. thereby improving the robustness of the student model.
>
> #### **Q3: Could the author explain more about the two processes of IB?**
> **A3**: Thank you for your comments and sorry for the confusion. IB expresses a trade-off in intermediate features $Z$ between relevant information for the prediction $Y$ and nuisance information about the input $X$.  The objective of IB can be formulated as follows:
> $
>     \max I(Z; Y) - \beta I(X; Z),
> $
> where $I$ denotes mutual information and $\beta$ controls the trade-off between the two terms.
>
> IB involves two processes. One is maximizing the first term ($ I(Z; Y)$), which means that ensures there is a strong correlation between the learned features and the target label. This correlation is relevant information.
> the other is maximizing the second term ($ -I(X; Z)$), which means ensuring that the learned features contain relevant information as much as possible and discard nuisance information.
>
> For the two processes in IB, we propose two distillation strategies to match the two optimization processes, respectively. First, we utilize robust soft label distillation to maximize the mutual information between latent features and output prediction. Second, we present an adaptive feature distillation that automatically transfers relevant knowledge from the teacher model to the student model, so that it can restrict the mutual information between the input and latent features. Finally, we integrate the two proposed optimizations into a final objective function.
>
> #### **Q4: Does the author have any ideas for improvement on this trade-off ?**
> **A4**: In our final optimization objective function (Eq. (16)), we introduce the $\alpha$ hyperparameter, which is a hyperparameter that can trade off the adversarial robustness and natural accuracy.
> We conduct ablation experiments to verify the trade-off. The results are shown in the following. When we set $\alpha = 0.9$, our method can achieve the best adversarial robustness.
>
> |  Alpha   |0.0  | 0.1   |0.2  |0.3  | 0.4   |0.5  |0.6   |0.7  | 0.8  |0.9   |1.0  |
> |  ----  | ----  | ----  | ----  | ----  | ----  | ----  | ----  | ----  | ----  | ----  | ----  |
> |  Natural   | 87.28  | 85.32  | 84.74  | 84.45  | 84.25   | 84.02  | 83.55   | 83.62  | 83.48  | 83.17  | 83.04  |
> |  AA   | 9.31  | 47.31  | 49.67  | 51.43  | 51.45  | 51.57  | 51.79   | 51.82  |  51.96  | 52.11  | 52.04  |
>
> In addition, in order to improve this trade-off, we train IBD with additional training data [2] to achieve better trade-off results, Our IBD can achieve 87.82% natural accuracy and 61.22% adversarial robustness under standard autoattack, which significantly improves the trade-off.
>
> Ref:
>
> [1] Adversarial Examples Are Not Bugs, They Are Features. NeruIPS 2019.
>
> [2]  Fixing Data Augmentation to Improve Adversarial Robustness NeruIPS 2021

---

> > ### Comment · Reviewer_wRTW · 2023-08-13
> > **Response to authors' rebuttal**
> >
> > Thanks to the authors' response which has addressed my main concerns, and I will raise my rating.

---

> > > ### Author Response · Authors · 2023-08-13
> > > **Response to Reviewer wRTW**
> > >
> > > Thank you for considering our response and providing feedback. We greatly appreciate your valuable suggestion and will incorporate additional explanations and experimental results into the revised manuscript to further strengthen our work. Thank you again for your time and input.

---

### Official Review · Reviewer_4SqU · 2023-07-07

**Soundness:** 3 good
**Presentation:** 3 good
**Contribution:** 3 good
**Rating:** 7
**Confidence:** 4

**Summary:**

This paper draws inspiration from prior studies that suggest robust models can offer strong prior information, thereby enhancing both the robustness and uncertainty of the model. Accordingly, we propose a new Information Bottleneck (IB) objective, which is designed to distil robustness in the context of a Variational Information Bottleneck (VIB).

**Strengths:**

Pros:

1. The proposed Information Bottleneck Distillation (IBD) method can significantly improve the robustness of Deep Neural Networks (DNNs), protecting them against most attacks such as the PGD-attack and AutoAttack.

2.  The IBD method optimizes the information bottleneck efficiently and effectively by maximizing mutual information between intermediate features and output prediction via soft label distillation, and restricting the mutual information between the input and intermediate features via adaptive feature distillation.

3. The adaptive feature distillation transfers appropriate knowledge from the teacher model to the student model, resulting in a more accurate estimation of the student feature distribution.

4. The method was extensively tested on various benchmark datasets, including CIFAR and ImageNet, demonstrating its effectiveness and robustness compared to state-of-the-art methods.



**Weaknesses:**

N/A



**Questions:**

N/A

---

> ### Author Rebuttal · Authors · 2023-08-10
>
> We thank you very much for your valuable and encouraging comments on our work!  Thanks!

---

### Official Review · Reviewer_rCCu · 2023-07-07

**Soundness:** 3 good
**Presentation:** 3 good
**Contribution:** 3 good
**Rating:** 5
**Confidence:** 3

**Summary:**

This paper proposes the Information Bottleneck Distillation (IBD) method to enhance adversarial robustness, derived from revisiting variational information bottleneck from the perspective of robustness distillation. IBD leverages two distillation strategies to perform the optimization processes of the information bottleneck, namely soft label distillation and adaptive feature distillation. The final experimental results show that the proposed method enhances the adversarial robustness against both the white- and black-box attacks.

**Strengths:**

1. This paper revisits variational information bottleneck from the perspective of robustness distillation, which utilizes the intermediate features extracted by a pre-trained teacher model to approximate $q(z)$.
2. The experimental results demonstrate that the proposed method improves the adversarial robustness against both the white- and black-box attacks.


**Weaknesses:**

1. This paper contains two hyperparameters, namely $\alpha$ and $\beta$. The authors should also analyze the impact of $\alpha$ on the proposed method.
2. In the experiment part of this paper, the classical adversarial training methods are SAT and TRADES. These benchmarks are somewhat old, and it is recommended to use newer training methods as baselines to boost the universality of the methods.
3. There are some minor typos in this paper, such as inconsistent presentation tenses, errors in the use of singular and plural (line 99), and the representation of the cross-entropy function in Eq.(2).


**Questions:**

Please refer to the weaknesses part.

**Limitations:**

The authors point out the potential limitations of the proposed method.

---

> ### Author Rebuttal · Authors · 2023-08-10
>
> Thanks for your valuable feedback! Here is our response to the concerned questions.
>
> #### **Q1: The impact of the hyperparameters $\alpha$**
>
> **A1**: Thanks for your comment. The $\alpha$ is a trade-off the adversarial robustness and natural accuracy.
> We conduct ablation experiments to verify the trade-off. The results are shown in the following table. When we set $\alpha = 0.9$, our method can achieve the best adversarial robustness.
>
> |  Alpha   |0.0  | 0.1   |0.2  |0.3  | 0.4   |0.5  |0.6   |0.7  | 0.8  |0.9   |1.0  |
> |  ----  | ----  | ----  | ----  | ----  | ----  | ----  | ----  | ----  | ----  | ----  | ----  |
> |  Natural   | 87.28  | 85.32  | 84.74  | 84.45  | 84.25   | 84.02  | 83.55   | 83.62  | 83.48  | 83.17  | 83.04  |
> |  AA   | 9.31  | 47.31  | 49.67  | 51.43  | 51.45  | 51.57  | 51.79   | 51.82  |  51.96  | 52.11  | 52.04  |
>
> #### **Q2: Some benchmarks are somewhat old**
> **A2**:Thanks for this fruitful advice.
> In order to verify the effectiveness of our method, we not only compare with some classical adversarial training methods (i.e., SAT and TRADES) but also compare with several state-of-the-art adversarial trained models on the robust benchmark[1] such as LBGAT[2]  (2021), LTD[3] (2021) and LAS-AT[4] (2022), all of which are published in recent years.
> As shown in the Table, we can observe that the proposed IBD improves the adversarial robustness by $\sim 1.2\%$.
>
> Furthermore, when combined with AWP [5], our IBD also surpasses the previously state-of-the-art models reported by the benchmark. where every small margin of improvement is significant. \textbf{Note} that our method does not use any additional datasets.
>
>  Method   |  WRN  |  Natural  | AA   |
>  ----  | ----  | ----  | ----  |
> SAT  | 34-10 | 84.92 | 53.42 |
> LBGAT  | 34-20 | 88.70 | 53.57  |
> TRADES  | 34-20 | 86.18 |  54.39 |
> LTD | 34-10 | 85.02 |  54.45  |
> IBD | 34-10 | 83.33 |  55.65  |
> TRADES + AWP | 34-10 | 85.26 |  56.17  |
> LASAT+ AWP | 34-10 | 84.98 | 56.26  |
> LTD+ AWP | 34-10 | 86.28 | 56.94  |
> IBD+ AWP | 34-10 | 85.21 | 57.18 |
>
> #### **Q3:  Some minor typos.**
> **A3**:  Thank you very much for kindly pointing this out.  We have corrected these typos and carefully checked the manuscript to ensure that it is typos-free.
>
> Ref:
>
> [1] Robustbench: a standardized adversarial robustness benchmark.2020
>
> [2] Learnable boundary guided adversarial training. ICCV 2021
>
> [3] Ltd: Low temperature distillation for robust adversarial training. 2021
>
> [4] Las-at: Adversarial training with learnable attack strategy. CVPR 2022
>
> [5] Adversarial weight perturbation helps robust generalization. NeurIPS 2020

---

> > ### Comment · Reviewer_rCCu · 2023-08-21
> >
> > I thank the authors for their comprehensive responses and new results. I decide to raise the score and hope the authors can incorporate the new results in the revision.

---

> > > ### Author Response · Authors · 2023-08-21
> > > **Response to Reviewer rCCu**
> > >
> > > Thank you for considering our response and providing feedback. We greatly appreciate your valuable suggestion and will incorporate the new experimental results into the revised manuscript to further strengthen our work. Thank you again for your time and feedback!

---

### Official Review · Reviewer_fxg6 · 2023-07-08

**Soundness:** 3 good
**Presentation:** 3 good
**Contribution:** 3 good
**Rating:** 5
**Confidence:** 4

**Summary:**

This paper aims to improve the adversarial robustness of deep neural networks. From the perspective of the Information Bottleneck, a knowledge distillation method is proposed. It makes use of intermediate features and logits from a robust teacher to get priors for guidance in training of the student model.

Experiments on CIFAR-10, CIFAR-100, and ImgaeNet are conducted. Obvious improvements are obtained when compared with previous methods.

**Strengths:**

1. The paper gives some insights into knowledge distillation from the perspective of Information Bottleneck.
2. The distillation method shows impressive results on model robustness.

**Weaknesses:**

1. From the perspective of knowledge distillation, feature-based methods have already been explored by previous methods, like [9], [ref1].
As the paper claims, the major difference between the proposed method and previous distillation methods is that an adversarially-trained robust model is used as the teacher.

 [ref1] Fitnets: Hints for thin deep nets. ICLR 2015.

2. It is interesting that the students can have stronger robustness than their teachers.
How would the performance of teachers affect that of student models?
Will the robustness of student models be enhanced when a more robust teacher model is deployed?




**Questions:**

See weakness

**Limitations:**

limitations are discussed in the supplementary file.

---

> ### Author Rebuttal · Authors · 2023-08-10
>
> Thanks for your valuable feedback! Here is our response to the concerned questions.
> #### **Q1: From the perspective of knowledge distillation, feature-based methods have already been explored by previous methods.**
> **A1**: Thanks for your comment. Our approach indeed resembles the feature-based distillation methods.
> However, we would like to highlight that the motivation behind our IBD and the optimization techniques is fundamentally different from those of the previous methods:
>
> 1) We utilize conditional variational inference to construct a lower bound to estimate the mutual information and reformat the IB principle by using the adversarial robustness as the prior for learning features, which is termed Information Bottleneck Distillation (IBD). The optimization of our IBD complies with the Information Bottleneck Principle, which makes the target model learns more relevant information and discard nuisance information.
>
> 2) From the perspective of robust distillation, previous adversarial robust distillation methods only optimize the output distribution of the model, while ignoring the feature information of the model.  Previous work [1] has shown that there are two kinds of features in the neural network: robust and non-robust features.  Therefore, we adopt the robust model as the teacher model and use the adversarial robustness as the prior for learning robust features. We propose an adaptive feature distillation strategy to automatically transfer robust features from the teacher model to the student model. thereby improving the robustness of the student model.
>
> #### **Q2: How would the performance of teachers affect that of student models?**
> **A2**: Thank you for your good questions. In our submitted manuscript, we only used one robust teacher network for fair experimental comparison. Following your comments, we conduct an ablation experiment by using different teacher models to verify the impact of the teacher's robustness on the performance of the student model. We conduct this experiment on CIFAR-10 with two student models: ResNet-18 and WideResNet-34-10, and five different teacher models which have different robustness. The results are shown in the Table. We can observe that different robust teacher models have a significant positive benefit on the student model. For the ResNet-18 student model, we find that the robustness of the student does not increase monotonically with that of the teacher.
> As the teacher model (WideResNet-34-20) becomes more complex, the robustness of the student model decreases, compared to WideResNet-34-10.  This may be due to the large gap in the architecture of the teacher model and the student model.
> This phenomenon is called **Robust saturation** [2].
> For the WideResNet-34-10 student model, we found that in most cases, the student’s robustness can surpass that of the teacher model.
> We think there are two reasons for this, one is that the performance of the teacher model is not very strong.
> The other is that the teacher model provides robust soft labels to alleviate overfitting and improve performance.
> Therefore, in most cases, it is expected that the student model exceeds the teacher model, but when the teacher model is strong enough, it is not easy for the student model to surpass the teacher model (e.g., WideResNet-76-10).
>
>  Teacher   | Natural  | AutoAtt   | Student  | Natural  | AutoAtt    | Student  | Natural   | AutoAtt   |
>  ----  | ----  | ----  | ----  | ----  | ----  | ----  | ----  | ----  |
> Resnet-18  | 84.09 | 48.71 | Resnet-18  | 83.74 | 50.52 | WRN-34-10  | 84.41 | 53.94 |
> Resnet-34  | 85.94 | 50.57 | Resnet-18  | 84.92 | 49.84 | WRN-34-10  | 85.79 | 54.17 |
> WRN-34-10  | 84.92 | 53.08 | Resnet-18  | 83.17 | 52.11 | WRN-34-10  | 84.21 | 55.65 |
> WRN-34-20  | 85.65 | 56.82 | Resnet-18  | 82.82 | 51.64 | WRN-34-10 | 84.73 | 55.71 |
> WRN-76-10  | 88.54 | 64.25 | Resnet-18  | 85.28 | 51.98 | WRN-34-10  | 86.61 | 57.12 |
>
> Ref:
>
> [1] Adversarial Examples Are Not Bugs, They Are Features. NeruIPS 2019.
>
> [2]  Revisiting adversarial robustness distillation: Robust soft labels make student better. ICCV 2021.

---

### Author Rebuttal · Authors · 2023-08-10

We thank all reviewers for their insightful and constructive comments on our work. During the rebuttal period, we carefully addressed all the comments and suggestions raised by all reviewers. We hope that our response has properly addressed the comments of the reviewers and that its overall contribution, quality and clarity are now significantly improved! Thanks all!

---

### Comment · Area_Chair_jj79 · 2023-08-19
**Please reply to author comments**

Dear reviewers of Submission1797,

    The deadline for the rebuttal/discussion period is approaching. Could you check the authors' comments and provide your feedback if you have not done? Thank you.

Your AC

---

### Decision · Program_Chairs · 2023-09-21

**Decision:**

Accept (poster)

**Comment:**

The paper received mixed ratings initially. The reviewers think the idea is technically sound and the results are impressive. However, there are some concerns regarding the details of the method, the setting of the experiments, and some analysis in the empirical study. After the rebuttal, the reviewers were mostly satisfied with the version the authors provided and raised their scores accordingly. AC is happy to see the paper accepted and the authors revise it incorporating the reviewers' feedback.